# Gamma Knife Radiosurgery for Trigeminal Neuralgia: A Comparison of Dose Protocols

**DOI:** 10.3390/brainsci9060134

**Published:** 2019-06-10

**Authors:** Warren Boling, Minwoo Song, Wendy Shih, Bengt Karlsson

**Affiliations:** 1Department of Neurosurgery, Loma Linda University, Loma Linda, CA 92354, USA; MINSong@llu.edu; 2School of Public Health, Loma Linda University, Loma Linda, CA 92354, USA; wshih@llu.edu; 3Department of Neurosurgery, National University Hospital, Singapore 119228, Singapore; nykuttram@yahoo.se

**Keywords:** trigeminal neuralgia, tic douloureux, radiosurgery, Gamma Knife

## Abstract

Purpose: A variety of treatment plans including an array of prescription doses have been used in radiosurgery treatment of trigeminal neuralgia (TN). However, despite a considerable experience in the radiosurgical treatment of TN, an ideal prescription dose that balances facial dysesthesia risk with pain relief durability has not been determined. Methods and Materials: This retrospective study of patients treated with radiosurgery for typical TN evaluates two treatment doses in relation to outcomes of pain freedom, bothersome facial numbness, and patient satisfaction with treatment. All patients were treated with radiosurgery for intractable and disabling TN. A treatment dose protocol change from 80 to 85 Gy provided an opportunity to compare two prescription doses. The variables evaluated were pain relief, treatment side-effect profile, and patient satisfaction. Results: Typical TN was treated with 80 Gy in 26 patients, and 85 Gy in 37 patients. A new face sensory disturbance was reported after 80 Gy in 16% and after 85 Gy in 27% (*p* = 0.4). Thirteen failed an 80 Gy dose whereas seven failed an 85 Gy dose. Kaplan–Meier analysis found that at 29 months 50% failed an 80 Gy treatment compared with 79% who had durable pain relief after 85 Gy treatment (*p* = 0.04). Conclusion: The 85 Gy dose for TN provided a more durable pain relief compared to the 80 Gy one without a significantly elevated occurrence of facial sensory disturbance.

## 1. Introduction

Trigeminal neuralgia (TN) is a chronic neuropathic pain condition that affects the regions of the face innervated by the trigeminal nerve. Typical TN causes severe and sudden volleys of shock-like facial pain that lasts a few seconds to a few minutes in the distribution of one or more divisions of the trigeminal nerve. When patients experience attacks of pain that come on repeatedly, the result is a pain condition that is disabling in nature. Typical TN will have a trigger in the trigeminal nerve division distribution the pain is felt in, which often leads to aversive behavior by the patient to avoid the pain. For example, a trigger inside the mouth may make eating so difficult that poor nutrition results.

Atypical forms of face pain have symptoms that are not shock-like in nature, do not follow the trigeminal nerve distribution, and/or do not have an associated trigger. Atypical face pain my certainly be disabling in nature, however, the treatment options available to typical TN patients are less likely to benefit individuals with atypical face pain. 

The etiology of typical TN in most cases is due to vascular compression at the nerve root entry zone as the trigeminal nerve exits the pons. In these cases, microvascular decompression surgery (MVD) in which a small sponge is placed as a cushion between the trigeminal nerve and the offending vessel has a high success rate in eliminating the TN pain. However, the associated risks and inconvenience related to the open surgical procedure of MVD lead many individuals with disabling pain, in particular more elderly and frail individuals, to seek out less-invasive treatment approaches such as radiosurgery and percutaneous rhizotomy. Both these approaches have a good success rate in eliminating the pain of TN, but pain recurrence is generally more common compared with MVD.

Lars Leksell introduced radiosurgery as a treatment of trigeminal neuralgia (TN) at the Karolinska Institute in the 1950s [1]. Although interest waned in treating TN with radiosurgery until the early 1990s when there was a substantial increase in the published literature concerning radiosurgery for TN. There is now over the past couple of decades considerable patient experience with radiosurgery in the treatment of TN with the vast majority of publications demonstrating benefit for the severe disabling facial pain [2], particularly in the elderly population who make up the majority of individuals with TN treated by radiosurgery [3].

Radiosurgery treatment variables that have potential to impact both long-term pain relief and occurrence of treatment side-effects include the radiation dose delivered to the nerve, volume or extent of the nerve treated, and anatomical localization of the treatment target. Of these variables, treatment dose has been most frequently studied. Yet, despite a substantial experience in the radiosurgery community, the optimum radiosurgery prescription dose for TN has not been determined. 

Herein, the authors present results from their patients with medically intractable TN treated using Gamma Knife radiosurgery (GKR). A retrospective comparison of two treatment dose plans of 80 and 85 Gy was analyzed for the treatment of typical TN. The variables assessed were pain relief, side-effect profile, and patient satisfaction.

## 2. Material and Methods

### 2.1. Patient Population

All patients were treated for medically intractable TN by the authors at a single institution over a seven year period. Patients (or a family member if the patient was deceased or unable to provide answers to a questionnaire) were contacted by phone or had a face-to-face interview to compete a study questionnaire at follow-up time points after treatment had been completed. The questionnaire inquired about timing of pain relief after GKR, quality and severity of pain, and side effects or complications related to GKR. The patient’s subjective assessment of GKR and treatment satisfaction was evaluated by asking two questions: “Are you pleased with GKR?” and “Would you have the procedure again?”.

### 2.2. GKR Treatment Plan

Gamma Knife Model 4c (Elekta, Stockholm, Sweden) was used for all treatments. The treatment plan centered the maximum dose on the root entry zone (REZ) of the proximal trigeminal nerve with the 30% isodose line just contacting the brainstem. Treatment was performed using a 4 mm collimated single shot. The earlier treatment plan was with a prescription maximum dose delivered to patients of 80 Gy. Later, the protocol was changed to a prescription maximum dose of 85 Gy in all subsequent patients. These two dose plans were analyzed and compared in treated patients for variables of pain relief, facial numbness, complications of treatment, and patient satisfaction (Figure 1).

### 2.3. Statistics

A Kaplan–Meier statistic analyzed the duration of pain freedom after GKR for low- and high-dose groups of patients. The Kaplan–Meier (K–M) survival distributions of the two dose groups were compared using the log rank test. A Fisher’s exact or chi square test compared categorical data, such as new onset facial numbness after GKR, patient satisfaction queries, and Barrow Neurological Institute pain intensity (BNI) at last follow-up [4]. *p* < 0.05 was considered significant.

## 3. Results

A total of 68 patients were treated with GKR for intractable TN over seven years. Three patients with multiple sclerosis (MS) (all in the 80 Gy group), and two with no available follow-up after treatment (one in each group) were excluded from the analysis. Therefore, in patients with typical TN and post-treatment follow-up, the treatment dose was 80 Gy in 26 individuals, and 85 Gy was delivered to 37 individuals. Mean patient age was 71 years. Twenty-seven were women. Fifteen patients had a procedure for TN prior to GKR (10 in the 80 Gy group, *p* = 0.6). The mean follow-up after GKR in pain free patients was 37 months (range 6–72 months) in individuals treated with 80 Gy and 26 months (range 6–52 months) in patients treated with 85 Gy. 

A new facial sensory disturbance was reported after an 80 Gy treatment dose in four patients (16%) and in 10 (27%) after an 85 Gy treatment dose (*p* = 0.4). Only one individual reported being bothered by numbness in the 80 Gy group and two reported that the sensory change was bothersome in the 85 Gy group. Patients answered “No” to either question of treatment satisfaction (Are you pleased with GKR? or Would you have the procedure again?) in 22% of the 85 Gy treatment group and in 44% of the 80 Gy treatment group (*p* = 0.09). 

BNI pain scores at last follow-up were evaluated in each patient. Pain freedom without medication (BNI score of I) was realized in eight individuals treated with 80 Gy and in 21 treated with 85 Gy, which represented more pain freedom without medication in the 85 Gy group (*p* = 0.04). In addition, significantly more patients who received 85 Gy treatment experienced an overall good result (BNI score I, II, and III) than patients treated with 80 Gy (29 with overall good results in the 85 Gy group versus 14 in the 80 Gy group, *p* = 0.04).

A survival curve was analyzed for both treatment groups using the Kaplan–Meier statistic (K–M). Recurrent severe pain despite medication or persisting severe pain after GKR was deemed a treatment failure. At the last follow-up, thirteen patients (52%) who received 80 Gy treatment dose failed GKR whereas seven patients (19%) failed GKR after 85 Gy. K–M analysis found at the 29 months time point that 50% of patients had failed 80 Gy GKR treatment, and at the same time point, 79% of patients had continued pain relief after receiving 85 GY treatment (Figure 2). The K–M analysis demonstrated a significant difference in the achievement of pain relief and durability of response at 29 months (*p* = 0.04).

Potential confounders of treatment success were evaluated to determine whether age, new numbness after GKR, surgery before GKR, or length of follow-up differed by GKR dosage (Figure 3). There were significant differences in age and prior surgery in the two GKR dosage groups. As a separate analysis, the effect of dosage on becoming pain free was analyzed while adjusting for potential confounders. In addition, only the effect of age and prior surgery were evaluated as possible confounders. For both analyses, the confounders were found to be not significantly contributing to the dose effect on pain freedom.

## 4. Discussion

Kondziolka et al. identified that patients who received a 70 Gy dose for treatment of TN fared better with improved pain relief compared to lower prescription doses, and complications were rare [5]. Also, the authors suggested in their patient population that higher treatment doses over 70 Gy yielded even better TN pain relief. However, unresolved is the ideal dose that provides the best-possible pain control balanced with the minimum associated radiation complications (Table 1). Additionally, there is no agreement in the published reports of dose escalation. Some authors have identified no benefit for TN or absence of improved pain relief from treatment with higher prescription doses ranging between 50 and 90 Gy [6,7,8,9,10,11,12]. For example, although an improvement in absolute pain relief was not found by Kim et al. with a higher radiation dose, these authors reported 85 Gy provided a more rapid response of pain relief after radiosurgery compared with the 80 Gy dose [13]. Consistent with our results, other authors have identified definite improved pain relief with dose escalation [14,15,16,17,18]. Longhi et al. evaluated radiosurgery for TN targeting the trigeminal nerve root entry zone. The authors identified higher doses in the 80–90 Gy range to be the most effective radiosurgery dose related to a pain free outcome that additionally had low risk of sensory disturbance, although this risk was elevated with the highest treatment dose [14]. 

Morbidini-Gaffney et al., in a follow-on study of patients reported in Alpert et al. [14], assessed patients treated with TN in order to evaluate efficacy of two versus one-isocenter treatment plans as well as an evaluation of radiation dose escalation [18]. The results were that two isocenters plus patients receiving greater than 85 Gy had a longer duration of good treatment response compared with lower treatment doses and a single isocenter. There were no identified facial dysesthesias although 11% reported mild facial numbness. In a conflicting report from Zhang et al. patients with refractory TN were treated with a maximum dose of 75–90 Gy using either one (*n* = 41) or two (*n* = 32) isocenters. The authors found no difference in pain relief or sensory disturbance with higher dose, but the patients with multiple-isocenter treatment plans did experience more numbness or paresthesia in the trigeminal distribution [12]. 

Smith et al. used linear accelerator-based radiosurgery to treat TN [17]. Over an initial time period, 28 patients received doses between 70 and 85 Gy. The subsequently treated 82 patients were prescribed a radiation dose of 90 Gy, with a treatment plan of the 30% isodose line touching the brainstem. The treatment plan was further modified later with the goal of increasing the dose at the root entry zone so that the 50% isodose line was tangential to the pons in 59 patients. Patients treated with 90 Gy had superior pain relief at one year follow-up and more rapid resolution of pain relief after treatment. There was no significant difference in facial numbness between the treatment groups. Young et al. reported over a five year follow-up after 90 Gy GKR treatment for TN that over 70% of patients were pain free with or without medication [22]. The authors concluded that higher-dose treatment was effective in treating TN, that pain relief was more likely in patients with facial numbness post treatment, and the authors commented that a 90 Gy prescription dose may be associated with an increase in bothersome sensory complications in comparison with lower treatment doses. Pollock et al. also identified increased numbness and dysesthesia in their patients treated with 90 Gy compared with lower doses [23]. The University of CaliforniaSan Diego TN treatment experience was reviewed by Taich et al., who identified elevated risk of bothersome facial numbness with treatment doses greater than 85 Gy [24]. Radiosurgery dose escalation was studied retrospectively in 870 TN patients by Kotecha et al. who analyzed patients divided into groups of delivered doses: ≤82, 83–86, and ≥90 Gy [19]. The investigators identified that dose escalation above 82 Gy resulted in improved pain relief, but with elevated risk of treatment-related facial numbness. However, facial numbness resulted in similar proportions in patients treated at prescription doses ≥83 Gy. Massager et al. [20] analyzed 358 patients with GKR-treated TN that targeted the anterior cisternal portion of the trigeminal nerve. The Brussels study divided patients into three different dosimetric treatment groups, which revealed rates of trigeminal numbness and pain relief closely related to the radiation dose delivered to the retrogasserian portion of the nerve. A similar relationship was reported by Villavicencio et al. [21]. This group treated TN using CyberKnife (Accuray Inc., Sunnyvale, CA, USA) radiosurgery, which resulted in better pain relief and increased hypesthesia rates in those patients receiving higher radiation dose with a longer nerve segment treated.

High-resolution MRI or CT myelography allows for the trigeminal nerve to be clearly delineated from its exit at the brainstem to Meckel’s cave. The imaging well-defined anatomy of the trigeminal nerve allows for accurate radiosurgery targeting at any point along its course in the subarachnoid cistern. The REZ has been the most common radiosurgery treatment site of TN likely because the REZ has a long history in neurosurgery as a lesioning target for a variety of pain conditions. However, in an effort to reduce the pontine radiation dose, some radiosurgery users have moved the target more anteriorly along the nerve [17]; although, few investigators have critically analyzed anatomical targeting differences related to outcome from treatment of TN (Table 2). Among the few authors who have critically evaluated a more anterior treatment plan, Matsuda et al. found no difference in pain freedom and a trend to more facial numbness using an anterior retrogasserian target in comparison to a REZ target, although the anterior target received a higher dose in their patients [16]. Park et al. identified no difference in pain freedom or facial numbness in patients treated with REZ versus an anterior retrogasserian target [25]. Rashid et al. treated with 90 Gy maximum dose to REZ and retrogasserian targets. Comparison of the two treatment groups identified improved pain freedom in the REZ target patients, and no difference in the development of new facial numbness [26]. Xu et al. retrospectively evaluated 141 patients with TN treated using GKR prescribed to a maximum dose of 80 Gy targeting either the REZ or the retrogasserian nerve. Their analysis revealed the REZ provided more durable pain relief with similar initial efficacy, but that facial numbness was more common in the REZ-treated patients [27]. Sharim et al., using linear accelerator-based radiosurgery, found no benefit in regard to face hypesthesia and no difference in pain relief rate with trigeminal nerve targets more anterior to the REZ [28]. Strategies such as anterior nerve targeting as well as techniques to correct dosimetry are effective in reduction of the brainstem total dose [29], and there is limited evidence that the risk of facial numbness may be reduced with a more anterior treatment target. This topic was reviewed by the International Radiosurgery Society Practice Guideline committee, which concluded that there is level II evidence that an anterior target reduces radiosurgery-related facial numbness with a similar benefit of pain reduction compared to REZ [30]. 

A few reports have described treating a greater nerve volume in an effort to improve pain-free results by employing an additional isocenter along the length of the nerve (Table 3). In a prospective randomized study, Flickinger et al. found an elevated risk of facial numbness with a two-isocenter treatment plan without improved pain relief [31]. However, in other published studies, Pollock et al. [8] and Alpert et al. [14] identified no relationship between nerve volume treatment and the development of facial numbness. Morbidini-Gaffney et al. [18], in a follow-on study from Alpert et al. [14], described in their patient population a significant positive correlation of multiple isocenters and higher treatment dose with pain freedom. They reported mild numbness in 11% of patients, although the hypesthesia risk associated with different treatment plans was not clear in the publication. Zhao et al. reported 247 patients who underwent a multi-isocenter GKR treatment of two adjacent 4 mm shots distributed along the trigeminal nerve with a maximum dose of 88 Gy [32]. In this group of patients treated, facial numbness occurred in 32.0%, of which 3.6% were identified as bothersome. Wolf et al. found in their TN patents treated with GKR no relationship of pain freedom with nerve length or nerve volume treated [33]. The authors did identify a relationship of better durability of pain relief at one year in patients with higher treatment dose delivered to smaller nerve volumes. Overall new facial numbness was reported in 23.6%, although only 3.6% experienced bothersome numbness. The preponderance of published studies analyzing multiple-isocenter treatment plans found no difference in pain-free outcome from more than one treatment shot, and that treatment-related facial numbness is not a consistently identified relationship with multi-shot treatment plans. However, the best evidence to date was from a single prospective randomized study [31], which found a significantly elevated risk of sensory changes related with a two-isocenter plan. 

Hypesthesia was rare in our patients (22% of all individuals treated for TN) with no difference between the two treatment doses. Most individuals treated by the authors who developed a sensory disturbance were not bothered by the numbness, typically patients commented that an improved quality of life resulting from resolution of the disabling pain was more relevant compared to the sensory change. Some patients also commented that a new facial numbness was an acceptable trade-off for pain relief. However, it should be noted that in three patients (21% of patients with a post-treatment sensory disturbance and 5% of all treated patients) bothersome sensory change was GKR treatment related, and the resulting dysesthesia in these individuals is reasonably regarded as a toxicity of the radiosurgery treatment. Seventy-eight percent of patients treated with 85 Gy maximum dose to the trigeminal nerve responded that they were pleased with the GKR and would choose the treatment again, which is similar to other reports of patient satisfaction. Debono et al. reported 86.5% patient satisfaction with LINAC radiosurgery of 90 Gy dose for trigeminal neuralgia [34]. 

Interventions that impact neural function are typically described as either ablative or modulatory. This has been discussed by other authors in regard to treatment of TN [35,36]. A seeming contradiction in discussions of radiosurgery for TN is that facial numbness resulting from radiosurgery is commonly labeled a treatment complication. A degree of facial numbness in the trigeminal branch of pain is a desired result of an ablative technique such as radiofrequency lesioning, which correlates positively with pain relief [37]. In fact, radiosurgery is more likely to yield pain relief in patients experiencing a facial sensory deficit, a fact that argues for an ablative mechanism [22,38]. However, neuromodulation is a better description of radiosurgery in the majority of patients who are able to realize pain relief without a sensory disturbance. Likewise, a dose–response relationship with hypesthesia has not been clearly demonstrated in studies of dose escalation (Table 1). In animal models of radiosurgery, a histopathological dose relationship has been demonstrated [39,40]. Minor changes become evident at about 80 Gy and necrosis is seen after the delivery of 100 Gy to the trigeminal nerve. In general, investigations up to now support elements of both neuromodulation and ablation as the physiological mechanism underlying radiosurgery treatment of TN, particularly at the radiation doses used to treat patients. 

The optimal radiosurgery treatment dose for TN should strike a balance between providing the most robust pain relief and the least possible risk of radiation-induced complications. The only radiation-related complication identified in our patient population was hemifacial dysesthesia, which was rare (4% in the 80 Gy treatment group and 5.4% in the 85 Gy group) and not significantly different between the two prescription doses. However, the patients who received the higher treatment dose of 85 Gy did realize a more robust treatment response compared with those who received the 80 Gy dose (79% versus 50% pain relief, respectively, at 29 months, K–M analysis, *p* = 0.04). The major limitations of our study are a retrospective analysis of outcome measures and a shorter follow-up period in the high-dose, 85 Gy, treatment group.

## 5. Conclusions

We identified in our patients with typical TN that a GKR prescription dose treatment of 85 Gy provided a longer lasting and more robust pain relief compared to a dose of 80 Gy. A facial sensory change occurred in 11% more of the individuals in the 85 Gy treatment group, although the difference did not reach statistical significance (*p* = 0.4). Dysesthesia or a bothersome facial numbness was rare, and these complications were equally distributed between the treatment groups. The majority of patients were satisfied with radiosurgery for TN and, as expected, patients treated with the more effective dose of 85 Gy demonstrated a trend of greater treatment satisfaction. However, there is no agreement in the published studies of efficacy of dose escalation for the radiosurgical treatment of TN, and determination of the ideal dose prescription that maximizes pain relief and minimizes dysesthesia will likely require a randomized prospective clinical trial.

## Figures and Tables

**Figure 1 brainsci-09-00134-f001:**
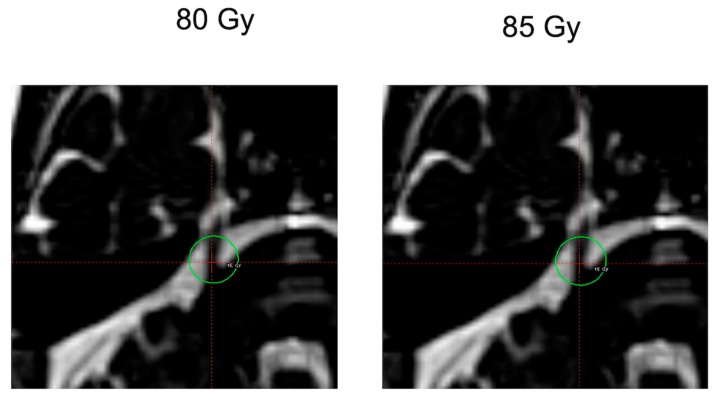
Images of typical corresponding dose plans with the maximum dose targeting the root entry zone (REZ). Green circle is the 15 Gy isodose line.

**Figure 2 brainsci-09-00134-f002:**
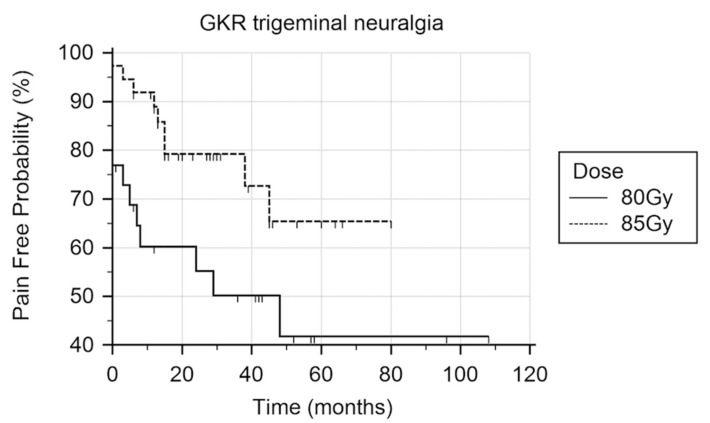
Kaplan–Meier analysis of Gamma Knife radiosurgery for trigeminal neuralgia using two treatment doses. Log rank test demonstrated improved durability and more patients with pain relief in the 85 Gy treated group (*p* = 0.04).

**Figure 3 brainsci-09-00134-f003:**
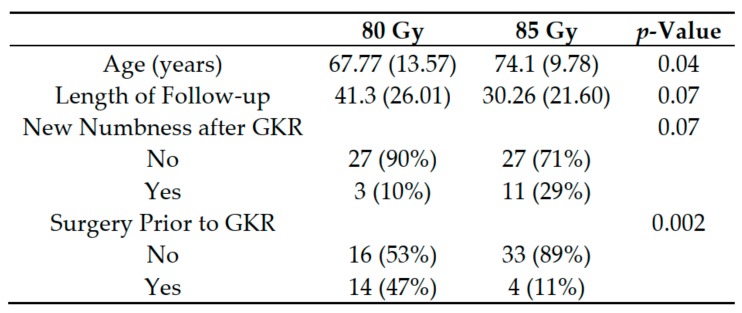
Analysis of potential confounders to pain freedom in the two dosage groups. Patient age and a procedure for trigeminal neuralgia prior to Gamma Knife radiosurgery (GKR) were both found to be significantly different between the dosage groups.

**Table 1 brainsci-09-00134-t001:** Studies of prescription dose comparison in radiosurgery of trigeminal neuralgia. * Linear accelerator radiosurgery, REZ = dorsal root entry zone, RG = retrogasserian.

Study	Max Dose to Nerve	Dose Related Pain Freedom	Facial Numbness Dose Related	Radiosurgery Target
Kondziolka et al. [5]	≤65 Gy ≥70 Gy	Yes ≥ 70 Gy *p* = 0.02	No relationship	REZ
Pollock et al. [8]	70 Gy 90 Gy	No difference	Yes. Numbness and dysesthesia	REZ
Alpert et al. [14]	≤80 Gy 85 Gy ≥90 Gy	Yes, with escalating doses *p* < 0.001	No relationship	REZ ±second shot 3–4 mm more distal
Sheehan et al. [9]	50–90 Gy	No difference	No relationship	REZ
Tawk et al. [10]	70, 80, or 90 Gy	No difference	Trend to dose relationship	REZ
Morbidini-Gaffney et al. [18]	<80 Gy 85 Gy >85 Gy	Yes < 85 Gy *p* < 0.001		REZ ±second shot 2–4 mm more distal
Régis et al. [11]	70–90 Gy	No difference	No relationship	7.5 mm anterior to pons
Fountas et al. [7]	75–85 Gy	No difference		REZ
Longhi et al. [15]	75–95 Gy >80 Gy	Yes > 80 Gy *p* = 0.008	>90 Gy increased numbness	REZ
Chen et al. * [6]	85 or 90 Gy	No difference		Cisternal nerve segment
Matsuda et al. [16]	80 or 90 Gy	No difference	Trend to more numbness in 90 Gy RG target	80 Gy REZ 90 Gy RG target
Kim et al. [13]	75 or 85 Gy	No difference	No relationship	REZ
Smith et al. * [17]	70 or 90 Gy	Yes, at one year	Trend to dose relationship	REZ
Zhang et al. [12]	75 or 90 Gy	No difference	No dose relationship	Cisternal portion of nerve with one or two isocenters
Kotecha et al. [19]	≤82, 83–86, or ≥90 Gy	Improved > 82 Gy	No dose relationship prescription doses ≥ 83 Gy	REZ
Massager et al. [20]	70–85 Gy, 90 Gy, or 90 Gy with shielding	No, trend to better pain freedom in higher dose	Yes. Numbness related with higher dose	Anterior cisternal nerve segment
Villavicencio et al. * [21]	Range of 50–80 Gy, median 75 Gy	Yes, related with longer nerve segment treated	Yes	REZ

**Table 2 brainsci-09-00134-t002:** Comparison studies of radiosurgery targets in the treatment of trigeminal neuralgia (TN). REZ = dorsal root entry zone.

Study	Radiosurgery Target and Max Dose to Nerve	Pain Freedom	Facial Numbness Target Related
Matsuda et al. [16]	80 Gy REZ 90 Gy retrogasserian target	No difference	Trend to more numbness in the anterior target patients
Park et al. [25]	80–90 Gy REZ or retrogasserian target	No difference	No relationship
Xu et al. [27]	80 Gy REZ or retrogasserian target	Similar initial pain relief. REZ more durable	More facial numbness in the REZ target patients
Rashid et al. [26]	90 Gy REZ or retrogasserian target	REZ target better pain control	No relationship

**Table 3 brainsci-09-00134-t003:** Radiosurgery studies of multi-isocenter treatment plans. REZ = dorsal root entry zone.

Study	Max Dose to Nerve	Pain Freedom	Facial Numbness	Radiosurgery Target
Flickinger et al. [31]	75 Gy one or two isocenters	No difference	Increased numbness with two isocenters	REZ ±second shot 2–4 mm distal to brainstem
Pollock et al. [23]	70–90 Gy one or two isocenters	Trend for longer length of treated nerve	No relation with nerve volume treated	Single anterior target ±second shot along nerve
Alpert et al. [14]	≤80, 85, or ≥90 Gy one or two isocenters	Improved with escalating dose, two-shot treatment received higher dose	No relation with nerve volume treated	REZ ±second shot 3–4 mm more distal
Morbidini-Gaffney et al. [18]	<80, 85, or >85 Gy one or two isocenters	Improved in ≥85 Gy dose and number of isocenters treated	11% mild numbness, unknown relation to number of isocenters	REZ ±second shot 2–4 mm more distal
Zhang et al. Neurol India, [12]	75–90 Gy, one or two isocenters	No difference detected in dose delivered or number of isocenters	Numbness or paresthesia increased with two isocenter treatment	Cisternal portion of trigeminal nerve
Zhao et al. [32]	88 Gy and two isocenters		32% numbness 3.6% bothersome numbness	Two adjacent 4 mm shots commencing at the REZ
Wolf et al. [33]	80–90 Gy initial GKR, 65–70 Gy repeat GKR, 80 Gy shorter-nerve treatment, 85 Gy longer-nerve treatment	No relationship to nerve length or volume treated More durable in higher dose to smaller volume	Numbness in 23.6%, bothersome in 3.6%	REZ

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
