# Peer review of "Gamma Knife Radiosurgery for Trigeminal Neuralgia: A Comparison of Dose Protocols"

_brainsci, 2019, doi:10.3390/brainsci9060134_

Round 1
Reviewer 1 Report
The authors described comparison of dose protocols for trigeminal neuralgia and dose of 85 Gy provided a more durable pain relief compared to 80 Gy without a significantly elevated occurrence of facial sensory disturbance. This is a retrospective study including relatively small number of patients. Dose issue regarding radiosurgery for trigeminal neuralgia was discussed with previous reports.
However, some issues need to be clarified.
Treatment plan
The authors described that the treatment plan centered the maximum dose on the root entry zone (REZ) of the proximal trigeminal nerve with the 30% isodose line just contacting the brain stem. There may be a difference in the position of center of maximum dose between 80 Gy and 85 Gy treatment protocols. Please provide a representative treatment plan of 80 Gy and 85 Gy protocol. Readers could understand the difference between these protocols.
Facial sensory disturbance
P3 L94
Result
‘A new facial sensory disturbance was reported after an 80 Gy treatment dose in 4 patients (16%) and in 10 (27%) after an 85 Gy treatment dose (p=0.4)’
Conclusion in abstract
‘Conclusion: 85 Gy dose for TN provided a more durable pain relief compared to 80 Gy without a significantly elevated occurrence of facial sensory disturbance’.
The result is not statistically significant. However, there is a difference in follow-up period between two treatment groups. The trigeminal dysfunction has latency period following gamma knife surgery. There is actually no difference in incidence of bothersome trigeminal dysfunction. A new facial sensory disturbance more frequently occurred in 85 Gy treatment group (11% difference, 27% vs. 16%), although the difference did not reach statistically significance.
I do not at all agree with the statement in the conclusion of the abstract. The authors’ conclusion is too strong.
Minor
P2 L60 ‘GY’ should be ‘Gy’.
P2 L63 ‘patient population’ should be ‘Patient population’
P2 L80 Need the definition of K-M.
Author Response
I thank Reviewer 1 for the thoughtful responses and requested edits.
I added a figure in the manuscript that shows the 80Gy and 85Gy plans side by side.
I agree with the reviewer there is a trend toward greater sensory disturbance in the 85Gy treatment group that did not reach statistical significance. Perhaps a greater number of treatment patients would have identified a difference. It is possible that longer follow-up could have identified more treated patients with a sensory change in both treatment groups. I was not able to reliably identify the latency to onset of sensory change in all the patients who described a new facial numbness. However, in 6 of the 14 individuals the numbness onset was found to be between 2 and 14 months after radiosurgery (mean 9.6months, median 11.5months). The perpetual challenge in gathering long term follow-up in this population of patients is that radiosurgery of trigeminal neuralgia is largely a treatment of elderly patients, often too frail for microvascular decompression surgery. Patient death and new onset of dementia were frequent challenges to follow-up in these patients. I reported in the manuscript the findings of no significant difference. However, I added a more verbal description of the data presented in the Conclusion: “A facial sensory change occurred in 11% more of the individuals in the 85 Gy treatment group, although the difference did not reach statistical significance (p=0.4).”
The minor issues were corrected in the manuscript.
Reviewer 2 Report
This is not randomized controlled study. This retrospective study uses historical control. Patients age, sex, duration and severity of neuralgia, past therapy and other known factor (ex, personality) were not controlled. Follow up of new group is short. Multifactorial regression analysis should be demonstrated to reveal the contribution of dose change.
Author Response
I thank Reviewer 2 for the thoughtful responses and requested edits.
I agree with the reviewer that this manuscript describes a retrospective review of the authors experience in using two treatment doses of radiosurgery for the treatment of trigeminal neuralgia. The Introduction concludes with the statement: “A retrospective comparison of two treatment dose plans of 80 GY and 85 GY was analyzed for the treatment of typical TN. The variables assessed were pain relief, side effect profile, and patient satisfaction.” The retrospective nature of the research is listed as a weakness at the conclusion of the Discussion. To be more clear as to the retrospective nature of the study, I added “retrospective” in the abstract Methods section. Additionally further description of the retrospective nature of the study was inserted in the Methods section in the body of the manuscript.
Unfortunately the study population size does not allow for controlling for variables such as age, sex, duration and severity of neuralgia, past therapy and other known factor (ex, personality). This analysis will need to be left for future studies.
I added to the manuscript a bio-statistician to analyze the data and answer to the reviewer’s comments about a multivariate analysis. Our biostatistician’s findings were that we are not powered to analyze the data in a multivariate analysis. She performed additional analysis of potential confounders which I added to the manuscript including an additional figure (Figure 3). Wendy Shi, PhD is now an author on the study. Her response is:
We thank the reviewer for the suggestion. Our current study is a retrospective evaluation of the effect of GKR on severity of pain period within a single institution. Unfortunately, our study is limited in sample size where only 68 subjects received GKR over a 7-year window. Hence, we are not powered to detect significant dosage effects in a multivariate regression analyses.
We did evaluate whether age, new numbness after GKR, surgery before GKR, length of follow-up differed by GKR dosage (see Table below) and there was significant differences in age and prior surgery in GKR by dosage. As separate analyses, we did evaluate the effect of dosage on pain free while adjusting for all the proposed confounders (by the reviewer) and another model with only age and prior surgery as the proposed confounders. For both models, the confounders were not found to be significant and were thus left out from the final model. Consequently, we presented the univariate analyses between dosage and pain free since all the proposed confounders were not significant in the multivariate models.
80 Gy | 85 Gy | p-value | |
Age | 67.77 (13.57) | 74.1 (9.78) | 0.04 |
Length of Stay | 41.3 (26.01) | 30.26 (21.60) | 0.07 |
New Numbness | 0.07 | ||
No | 27 (90%) | 27 (71%) | |
Yes | 3 (10%) | 11 (29%) | |
Prior Surgery GKR | 0.002 | ||
No | 16 (53%) | 33 (89%) | |
Yes | 14 (47%) | 4 (11%) | |
Round 2
Reviewer 1 Report
The manuscript has been revised well.
Reviewer 2 Report
I found some improvements of the manuscript.